# ΔFBA—Predicting metabolic flux alterations using genome-scale metabolic models and differential transcriptomic data

**Sudharshan Ravi**[1,2], **Rudiyanto Gunawan**[1]*

**1** Department of Chemical and Biological Engineering, University at Buffalo-SUNY, Buffalo, New York, United States of America, **2** Institute for Chemical and Bioengineering, ETH Zurich, Zurich, Switzerland

* rgunawan@buffalo.edu

**Data Availability Statement:** All relevant data and codes are available at https://github.com/CABSEL/DeltaFBA.

## Abstract

Genome-scale metabolic models (GEMs) provide a powerful framework for simulating the entire set of biochemical reactions in a cell using a constraint-based modeling strategy called flux balance analysis (FBA). FBA relies on an assumed metabolic objective for generating metabolic fluxes using GEMs. But, the most appropriate metabolic objective is not always obvious for a given condition and is likely context-specific, which often complicate the estimation of metabolic flux alterations between conditions. Here, we propose a new method, called ΔFBA (deltaFBA), that integrates differential gene expression data to evaluate directly metabolic flux differences between two conditions. Notably, ΔFBA does not require specifying the cellular objective. Rather, ΔFBA seeks to maximize the consistency and minimize inconsistency between the predicted flux differences and differential gene expression. We showcased the performance of ΔFBA through several case studies involving the prediction of metabolic alterations caused by genetic and environmental perturbations in *Escherichia coli* and caused by Type-2 diabetes in human muscle. Importantly, in comparison to existing methods, ΔFBA gives a more accurate prediction of flux differences.

## Author summary

Metabolic alterations are often used as hallmarks of observable phenotypes. In this regard, reconstructed genome-scale metabolic models (GEMs) provide a rich and computable representation of the entire set of biochemical reactions in a cell. However, the performance of analytical tools for predicting metabolic reaction rates or fluxes using GEMs is sensitive to the assumed metabolic objective that is often unknown and likely context-specific. Here, we propose a novel method called ΔFBA that combines differential gene expression data and GEMs to evaluate differences in the metabolic fluxes between two conditions (perturbation vs. control) without the need for specifying a metabolic objective. In our demonstration, ΔFBA outperformed other existing methods in predicting metabolic flux alterations.

**Funding:** SR and RG received funding from Swiss National Science Foundation (grant number 163390 and 176279; www.snsf.ch). The funders had no role in study design, data collection and analysis, decision to publish, or preparation of the manuscript.

**Competing interests:** The authors have declared that no competing interests exist.

This is a *PLOS Computational Biology* Methods paper.

## Introduction

In the post-genomic era, there has been intense efforts directed toward the reconstruction of genome-scale models of cellular networks. An important portion of these efforts focuses on metabolic networks due to the significance of cellular metabolism for understanding diseases such as cancer [1–4] as well as for metabolic engineering applications in biomanufacturing [5]. Recent advances in high-throughput sequencing technologies, gene functional annotation, and metabolic pathway databases, and developments of algorithms for mapping gene-protein-reaction (GPR) associations and identifying missing metabolic reactions systematically (gap-filling), have enabled the reconstruction of thousands of genome-scale metabolic models (GEMs), from single cell organisms to human [6,7]. A GEM provides GPR associations that encompass the set of metabolites and metabolic reactions in an organism as prescribed by its genome. Concurrent with these developments is the creation of efficacious algorithms that use GEMs to predict intracellular metabolic fluxes–the rates of metabolic reactions–and how these fluxes vary under different environmental, genetic, and disease conditions [8–10].

A prominent class of algorithms based on a constrained-based modeling technique called flux balance analysis (FBA) have flourished due to its ease of formulation and flexibility. FBA uses the stoichiometric coefficients of the metabolic reactions in a GEM, an assumed cellular objective such as maximization of biomass production, and experimental data on metabolic capabilities and constraints of the cells, to predict metabolic fluxes [11]. Although FBA is effective in handling large networks and predicting cell behavior in many metabolic engineering studies [12–15], considerable uncertainty still remains about the appropriate choice of cellular objective for different conditions and cell types, a choice that typically requires expert knowledge of the cells and their phenotype in a given condition. Such an issue is particularly prominent for complex organisms such as human. Moreover, multiple equivalent flux solutions exist that give the same cellular objective value [16]. Not to mention, the standard FBA often produces biologically unrealistic flux solutions [17,18].

Driven by the increasing ease and availability of whole-genome omics profiling data, a multitude of FBA-based algorithms have been developed to incorporate omics datasets to create context-specific metabolic networks and to improve flux prediction accuracy [19–26]. Several of these methods, such as GIMME (Gene Inactivity Moderated by Metabolism and Expression) [20], iMAT (integrative Metabolic Analysis Tools) [21], and MADE (Metabolic Adjustment by Differential Expression) [22], are based on maximizing the consistency between the predicted flux distribution and the mRNA transcript abundance of metabolic genes, where the higher the transcript level of an enzyme, the larger should the flux of the corresponding reactions. Recent methods use data of mRNA transcript abundance for setting the bounds on reaction fluxes, e.g. E-Flux [23], or in the biological objective function, e.g. Lee et al. [24] and RELATCH (RELATive Change) [25]. Meanwhile, others like GX-FBA [26] determine fluxes in a perturbed state using differential gene expression and FBA flux prediction for the control (reference) state. Interestingly, a systematic evaluation of different FBA methods that incorporate gene expression data revealed a surprisingly poorer performance of these methods when compared to FBA with growth maximization and parsimony criteria, referred to parsimonious FBA (pFBA) [27]. More recently, ME-model [28] and GECKO [29] combine FBA with an explicit modeling of enzyme/protein expression and thus, are able to directly account for protein abundance. Thermodynamics constraints have also been integrated with the FBA to eliminate thermodynamically infeasible fluxes, and at the same time enable the integration of

metabolite concentration data, as done in recent methods such as ETFL [30]. All of the afore-mentioned methods, however, revolve around using omics data to predict metabolic fluxes for a given condition. But, many a times we are interested in the metabolic alterations caused by a perturbation or a change in intra/extracellular conditions.

Thus far, only a handful of methods focus on using differential expression data between two conditions (e.g., perturbation vs. control) to predict metabolic alterations directly, which is a particular focus of our study. The method Relative Expression and Metabolic Integration (REMI) [31] used differential expression of transcriptome and metabolome to estimate meta-bolic flux profiles in *Escherichia coli* under varying dilution and genetic perturbations. The method relies on maximizing the agreement between the fold-changes of metabolic fluxes and the fold-changes of enzyme expressions between two conditions. The metabolome data, if available, are used to determine the flux directionality using reaction thermodynamics. Among the alternative flux solutions, the L1-norm minimal solution is adopted to give a repre-sentative flux distribution. Another method by Zhu *et al.* [32] employed a softer definition when assessing consistency between the metabolic fluxes and enzyme differential expressions, where only the sign of the differences needs to agree. The method provides a qualitative deter-mination of metabolic flux changes by determining the maximum and minimum flux through each reaction in the GEM. Both of the above methods generate metabolic flux predictions for each of the conditions in comparison. Also, like the standard FBA, both methods require an assumption on the cell's metabolic objective. Generally, model prediction inaccuracy is ampli-fied when evaluating the differences between two model predictions. Another related method MOOMIN [33] uses a Bayesian approach to integrate differential gene expression profiles with GEMs to predict the qualitative change in the metabolic fluxes—increased, decreased or no change.

In this work, we developed ΔFBA (deltaFBA) for predicting the metabolic flux difference given a GEM and differential transcriptomic data between two conditions. ΔFBA relies on a constrained-based model that governs the balance of flux difference in the GEM, while maxi-mizing the consistency and minimizing inconsistency between the flux alterations and the gene expression changes. ΔFBA is developed as a MATLAB package that works seamlessly with the COnstraint-Based Reconstruction and Analysis (COBRA) toolbox [34]. We applied the ΔFBA to analyze the metabolic changes of *Escherichia coli* in response to environmental and genetic perturbations using data from the studies of Ishii *et al.* [35] and Gerosa *et al.* [36]. We compared the performance of ΔFBA in evaluating flux differences between conditions to that of REMI and eight FBA methods, including parsimonious FBA (pFBA) [19], GIMME [20], iMAT [21], MADE [22], E-Flux [23], Lee et al. [24], RELATCH [25], and GX-FBA [26]. We also demonstrated the application of ΔFBA to a human GEM, specifically evaluating the metabolic alterations associated with type-2-diabetes in skeletal muscle using myocyte-specific GEM [37].

## Materials and methods

### Method formulation

ΔFBA generates a prediction for metabolic flux differences between a pair of conditions, for example, treated vs. untreated or mutant vs. wild-type strains. In the following, we use the superscript C to denote the control (reference) condition and P to denote the perturbed condi-tion. In the standard FBA, writing mass balance around every metabolite and applying the steady state assumption give a linear equation $Sv = 0$, where $S \in \mathbb{R}^{m \times n}$ denotes the stoichiomet-ric matrix for $m$ metabolites that are involved in $n$ metabolic reactions and transports in the GEM and $v \in \mathbb{R}^n$ denotes the vector of $n$ fluxes (rates). In ΔFBA, the steady state flux balance

is assumed for each condition, and consequently, the flux difference $\Delta v = (v^P - v^C)$ satisfies the following balance equation:

$$S\Delta v = S(v^P - v^C) = Sv^P - Sv^C = 0$$

where $v^C \in \mathbb{R}^n$ and $v^P \in \mathbb{R}^n$ denote the vectors of metabolic fluxes in C and P, respectively, and $\Delta v \in \mathbb{R}^n$ denotes the vector of metabolic flux differences. The prediction of $\Delta v$ is based on maximizing the consistency while minimizing the inconsistency between the flux changes $\Delta v$ and the differential reaction expressions, constrained by among other things, the flux balance equation above. The following constrained mixed integer linear programming (MILP) gives the main formulation for ΔFBA:

$$\max_{z^U, z^D} \Phi = \max_{z^U, z^D} \sum_{i \in \mathbf{R}^U} w_i^U (z_i^U - z_i^D) + \sum_{j \in \mathbf{R}^D} w_i^D (z_j^D - z_j^U) \tag{1}$$

subject to:

$$S\Delta v = 0 \tag{2}$$

$$\Delta v_{min} \leq \Delta v \leq \Delta v_{max} \tag{3}$$

$$\Delta v - Mz^U \geq \mu - M\mathbf{1} \tag{4}$$

$$\Delta v - Mz^U \leq \mu \tag{5}$$

$$\Delta v + Mz^D \leq -\eta + M\mathbf{1} \tag{6}$$

$$\Delta v + Mz^D \geq -\eta \tag{7}$$

$$\Delta v + Mz^0 \leq M\mathbf{1} \tag{8}$$

$$\Delta v - Mz^0 \geq -M\mathbf{1} \tag{9}$$

$$z_k^0 + z_k^U \geq +z_{k'}^D \tag{10}$$

$$z_k^0 + z_k^D \geq +z_{k'}^U \tag{11}$$

Eqs (2) and (3) ensure that the flux difference $\Delta v$ satisfy the flux balance equation while staying within acceptable lower and upper bounds. The constrained MILP produces the optimal binary vectors $z^U \in \mathbb{Z}^n$ and $z^D \in \mathbb{Z}^n$ that maximize the consistency and minimize the inconsistency between the flux differences and the differential gene expressions. When $z_i^U = 1$, $\Delta v_i$ takes a positive value beyond the threshold $\mu_i$, as specified by the constraints in Eqs (4) and (5). When $z_i^D = 1$, $\Delta v_i$ takes a negative value beyond a threshold $\eta_i$, as specified by Eqs (6) and (7). Clearly, $z_i^U$ and $z_i^D$ cannot simultaneously be equal to 1. Meanwhile, the binary variable $z^0 \in \mathbb{Z}^n$ is used to force certain user-selected reactions, if any, to have zero flux change value, as specified by Eqs (8) and (9). Note that all reversible reactions in the GEM are written as two separate irreversible reactions, whose indices are denoted by $k$ and $k'$, the former for the forward and the latter for the backward direction. For all half pairs of reversible reactions, Eqs (10) and (11) ensure that the forward and reverse reactions are prevented to simultaneously have non-zero values, which is done to reduce degeneracy of the flux change solution $\Delta v$.

Finally, the constant $M$ in Eqs (4)–(9) should be set to a large value (default = $10^5$), following the Big M method in linear programming [38].

The set of upregulated reactions $\mathbf{R}^U$ and downregulated reactions $\mathbf{R}^D$ are user-defined inputs. More specifically, the sets $\mathbf{R}^U$ and $\mathbf{R}^D$ include indices of reactions with significant increase and decrease in gene expression between the perturbed condition and the control, respectively. The non-negative weighting coefficients $w_i^U \in \mathbb{R}$ and $w_j^D \in \mathbb{R}$ (default value = 1) in the objective function allow users to prioritize certain reactions for consistency among those in the sets $\mathbf{R}^U$ and $\mathbf{R}^D$, respectively. For example, the reaction corresponding to a gene deletion should be assigned a high $w_j^D$ to force the corresponding flux change to be negative. The upper and lower bounds for the flux differences in Eq (3) are also user-defined parameters that can be set based on experimental data (e.g., the difference of experimentally determined biomass production or growth rates) or based on the flux bounds from each condition. For the latter, given the lower and upper flux bounds for the $i$-th flux in the perturbed ($v_{min,i}^P \in \mathbb{R}$ and $v_{max,i}^P \in \mathbb{R}$, respectively) and the control condition ($v_{min,i}^C \in \mathbb{R}$ and $v_{max,i}^C \in \mathbb{R}$, respectively), the bounds for the flux difference can be set as follows:

$$\Delta v_{min,i} = \left( v_{min,i}^P - v_{max,i}^C \right) \tag{12}$$

$$\Delta v_{max,i} = \left( v_{max,i}^P - v_{min,i}^C \right) \tag{13}$$

Finally, the thresholds $\mu_i$ and $\eta_i$ for the positive and negative flux differences, respectively, are user-defined parameters. In the case studies, we used the same constant threshold value $\varepsilon$ (default = 0.1% of the largest flux bound magnitude in the two conditions). These thresholds serve as a lower (upper) bound for which a positive (negative) flux difference is deemed to be upregulated (downregulated).

Given the degrees of freedom in GEMs for $\Delta \boldsymbol{v}$, many equivalent optimal solutions often exist that give the same objective function value $\Phi^*$ as specified in Eq (1). By assuming parsimony for $\Delta \boldsymbol{v}$—that is, $\Delta \boldsymbol{v}$ is minimal between the perturbed and control condition—a two-step optimization procedure is implemented in ΔFBA. The first step is to maximize consistency with gene expression changes as prescribed in Eqs (1)–(11) to determine the maximum objective function value, denoted by $\Phi^*$. The second step is to produce an L2 norm minimal solution for $\Delta \boldsymbol{v}$, as follows:

$$\min_{\Delta \boldsymbol{v}, \boldsymbol{z}^U, \boldsymbol{z}^D} \| \Delta \boldsymbol{v}( \boldsymbol{z}^U, \ \boldsymbol{z}^D) \|_2^2 \tag{14}$$

subject to the same constraints in Eqs (2)–(11) while achieving the same level of consistency $\Phi^*$, implemented by the following additional constraint:

$$\sum_{i \in \mathbf{R}^U} w_i^U (z_i^U - z_i^D) + \sum_{j \in \mathbf{R}^D} w_i^D (z_j^D - z_j^U) = \Phi^* \tag{15}$$

The L2 minimization is based on the premise that the flux differences should be small between the conditions, which is similar to the method called Minimization of Metabolic Adjustment (MOMA) [39]. An alternative to L2-norm minimization is L1-norm minimization, which is analogous to maximizing sparsity of $\Delta \boldsymbol{v}$. The L1-norm minimization was previously used in parsimonious FBA (pFBA) method [19], but such an approach often still leads to multiple degenerate solutions. On the other hand, the L2-norm minimization will produce a unique solution. However, the mixed integer quadratic optimization that is required to find the minimum L2-norm solution may have high computational requirement.

ΔFBA is available as MATLAB scripts and are compatible with the COBRA toolbox [34]. ΔFBA requires Gurobi optimizer (http://www.gurobi.com) as a pre-requisite. ΔFBA has been tested on a Windows PC using a 6-core Intel Xeon (2146G) Processor with 16 GB RAM.

## Gene-protein-reaction mapping

For mapping fold-change gene expression data to fold-change reaction expression, we utilized the gene-protein-reaction (GPR) associations that are built into the GEM. These associations do not follow a one-to-one relationship since metabolic enzymes include isozymes (multiple enzymes mapping to the same reaction), promiscuous enzymes (a single enzyme participating in multiple reactions), and enzyme complexes (multiple genes required for an enzyme). Here, we used the Min/Max GPR rule [21,40,41]. The fold-change gene expression is first mapped to fold-change protein/enzyme expression. When multiple genes are required to form an enzyme complex, the fold-change enzyme expression is set to the minimum fold-change expression of the participating genes. Otherwise, the fold-change enzyme expression is equal to the fold-change gene expression. Then, the fold-change enzyme expression is mapped to the fold-change reaction expression. Here, when isozymes are involved in a reaction, the fold-change reaction expression is set to the maximum fold-change expression of the isoenzymes. Otherwise, the fold-change reaction expression is equal to the fold-change enzyme expression. Finally, based on the reaction expression, we prescribed the set of upregulated reactions $\mathbf{R}^U$ and the set of downregulated reactions $\mathbf{R}^D$ based on the fold-change reaction expression.

## Case studies: Data and implementation

The first case study involved the response of *E. coli*'s metabolism to genetic (single-gene deletions) and environmental perturbations (dilution rates) performed by Ishii *et al.* [35]. The study provided $^{13}$C-based flux data and RT-PCR mRNA abundances for the central carbon metabolism, pentose phosphate pathway (PPP), and the tricarboxylic acid (TCA) cycle for wild-type K12 *E. coli* culture in chemostat under different dilution rates (0.1, 0.2, 0.4, 0.5, and 0.7 hours$^{-1}$) and for 24 single-gene perturbations along the glycolysis and PPP [35]. The global transcriptional response was only captured for 5 of the 24 single-gene deletions (*pgm*, *pgi*, *gapC*, *zwf* and *rpe*) and two of the 4 dilution conditions (0.5 and 0.7 hours$^{-1}$). The differential (fold-change) gene expression levels were computed with respect to the control condition that was set to be wild-type K12 *E. coli* cultured at a dilution rate of 0.2 h$^{-1}$. The differential (fold-change) reaction expressions were subsequently evaluated based on the fold-change gene expression using the GPR Max/Min rule in the COBRA toolbox (MATLAB) [40]. For samples with only RT-PCR mRNA abundance data, the set of up- and downregulated reactions included all reactions with fold-change reaction expressions higher than 1 and those with fold-change lower than 1, respectively. In the additional analyses for samples with whole-genome transcriptome data, the set of up- and downregulated reactions were taken from the top and bottom 5$^{th}$ percentile of the differential reaction expressions. The differences of the measured cell specific glucose uptake rates between perturbed and control experiments were used as constraints. ΔFBA was applied using the two-step optimization with the L2 norm minimization, as described above.

The second case study came from a study of *E. coli* growth on 8 different carbon sources performed by Gerosa *et al.* [36]. Unprocessed global transcriptomic data were obtained from ArrayExpress (E-MTAB-3392), and differential expression analyses between every pair of carbon sources were evaluated using the *Limma* package in R [42]. We only included the set of genes with significance fold-change expression at FDR < 0.05. As before, the fold-change reaction expressions were computed based on fold-change in the global gene expression using the

Max-Min GPR rule using COBRA toolbox [40]. The up- and downregulated set of reactions were taken from the top and bottom 5th percentile of the differential reaction expressions. In addition, cell culture data on specific growth rates were used to compute the bounds for flux difference for biomass production rate. The uptake rates of the carbon source changes were also incorporated as constraints. We implemented the two-step optimization of ΔFBA using L2 norm minimization.

The third case study came from two studies of skeletal muscle tissue metabolism in type-2 diabetes (T2D) patients by van Tienen *et al.* [43] and Jin *et al.* [44]. The microarray gene expression datasets were obtained from GEO (GSE19420 [43] and GSE25462 [44,45], respectively) and the differential (fold-change) expression of genes for each dataset were computed using the *Limma* package in R [42]. We only included the set of genes with significance fold-change expression at FDR < 0.05. The fold-change reaction expressions were computed based on the differential gene expression using the Max/Min GPR rule [40]. In the absence of additional constraints in the form of exchange fluxes or growth characteristics, we set the up- and downregulated reactions from the top and bottom 25th percentile in differential reaction expressions, rather than the 5th percentile threshold used in *E. coli* case studies above, so as to incorporate more differentially expressed transcripts. We implemented an L1-norm minimization in the second step of ΔFBA to reduce computational complexity (time) due to the large number of constraints associated with the differential reaction expressions.

## Implementation of comparative methods

Among the comparative methods in this work, the method Relative Expression and Metabolomic Integrations (REMI) was specifically developed for predicting individual flux distributions of a pair of conditions ($v^P$ and $v^C$) using multi-omics dataset, and thus more comparable to ΔFBA. The toolbox was downloaded from https://github.com/EP-LCSB/remi. The differential gene expressions in each case study were obtained as described above. The mapping from differential gene expression to the corresponding reaction expressions were done using the procedure detailed in REMI [31]. Briefly, the authors followed the implementation of Fang *et al.* [34] to translate gene expression ratios to obtain reaction expression ratios. When several enzyme subunits are required for a reaction, a geometric mean of expression ratios is chosen to represent the reaction ratio. In the case where multiple isozymes catalyze a reaction, the arithmetic mean of the individual expression ratios of the isozymes is used for the reaction ratio. The set of up- and down-regulated reactions $\mathbf{R}^U$ and $\mathbf{R}^D$ were taken from the computed differential reaction expressions as in ΔFBA implementation. Unlike ΔFBA, REMI produces solutions for the metabolic fluxes of perturbed $v^P$ and control condition $v^C$. For comparison, we evaluated the flux change predicted by REMI by taking the difference: $\Delta v = v^P - v^C$.

We also considered 8 additional FBA methods with transcriptome data integration, including parsimonious FBA (pFBA) [19], GIMME [20], iMAT [21], MADE [22], E-Flux [23], Lee *et al.* [24], RELATCH [25], and GX-FBA [26]. The implementation of each of these 8 methods was described in a previous systematic comparison [27]. For performance evaluation, we again evaluated the differences of flux predictions: $\Delta v = v^P - v^C$.

## Performance evaluation

The agreement between the predicted flux changes $\Delta v^*$ and the ground truth [13]C-based flux difference $\Delta v^M$ was assessed by using two accuracy metrics: uncentered Pearson correlation coefficient and normalized root mean square error (NRMSE). The uncentered Pearson

correlation coefficient $\rho$ was computed as follows

$$\rho = \frac{\Delta \boldsymbol{v}^M \cdot \Delta \boldsymbol{v}^*}{\|\Delta \boldsymbol{v}^M\|_2 \|\Delta \boldsymbol{v}^*\|_2} \tag{16}$$

Meanwhile, the NRMSE was according to the following equation—using *tdStats* package in R:

$$NRMSE = \frac{1}{\Delta v_{max}^M - \Delta v_{min}^M} \sqrt{\frac{\|\Delta \boldsymbol{v}^M - \Delta \boldsymbol{v}^*\|_2^2}{n_M}} \tag{17}$$

where $n_M$ is the number of measured fluxes. Besides the quantitative agreement in flux changes, we also evaluated the qualitative agreement by comparing the signs of the flux changes between experimental measurements and predictions. To this end, we discretized the measured and predicted flux changes into +1, 0, and −1, to describe upregulated, no change, and downregulated reactions, respectively. The agreement in the direction of the flux changes was evaluated as the number of correct sign predictions divided by the total number of fluxes.

## Metabolic subsystem enrichment analysis

The flux differences obtained from applying ΔFBA were first filtered according to the directionality of their change. The significantly altered fluxes ($|\Delta v_i| > \varepsilon$) were grouped based on the subsystem to which the fluxes belong. A Fisher exact test (*fisher.test* function in the R-package) was used in determining over-represented subsystems in upregulated (positive change) and downregulated (negative change) fluxes. The statistical significance *p*-values were corrected for multiple hypothesis testing using the *p.adjust* function in R.

## Results

### *Escherichia coli* response to genetic and environmental variations

Ishii *et al.* [35] studied the robustness of *E. coli* K12 metabolism in chemostat in response to changes in dilution rates and to gene deletions. The study generated multi-omics data, including transcriptomic, proteomic, metabolomic, and $^{13}$C metabolic fluxes, and demonstrated the remarkable ability *of E.* coli to reroute its metabolic fluxes to maintain metabolic homeostasis in response to environmental and genetic perturbations. But, only a small fraction of variation in the measured flux ratios can be explained by the fold-change in reaction expressions, as indicated by the low coefficient of determinations $R^2$ ($R^2 = 0.088 \pm 0.059$). The low agreement between reaction expressions and metabolic fluxes suggests that metabolic fluxes are only weakly controlled by the gene expression. The formulation of ΔFBA is driven by two main assumptions: (1) first and foremost, that metabolic flux differences are balanced—an assumption that follows directly from steady-state flux balances in the control and perturbed conditions, and in addition (2) that the flux differences should be maximally consistent with the gene expression changes. Note that ΔFBA allows for inconsistency between differential gene expression and flux difference—for example, the gene expression is downregulated, but the flux difference is positive—but such inconsistency is kept low through a constrained MILP optimization.

We applied ΔFBA using *E. coli*'s *iJO1366* GEM to predict the metabolic flux shifts from the control condition (wild-type K12 at 0.2 hour$^{-1}$ dilution rate), caused by alterations in dilution rates (0.1, 0.4, 0.5, and 0.7 hours$^{-1}$) and by 24 single-gene deletions (*galM, glk, pgm, pgi, pfkA, pfkB, fbp, fbaB, gapC, gpmA, gpmB, pykA, pykF, ppsA, zwf, pgl, gnd, rpe, rpiA, rpiB, tktA, tktB, talA,* and *talB*), one condition at a time. For each single-gene deletion experiment, the knocked-out reaction was included in the set $\mathbf{R}^D$ and was assigned weighting $w_j^D = 10$, keeping

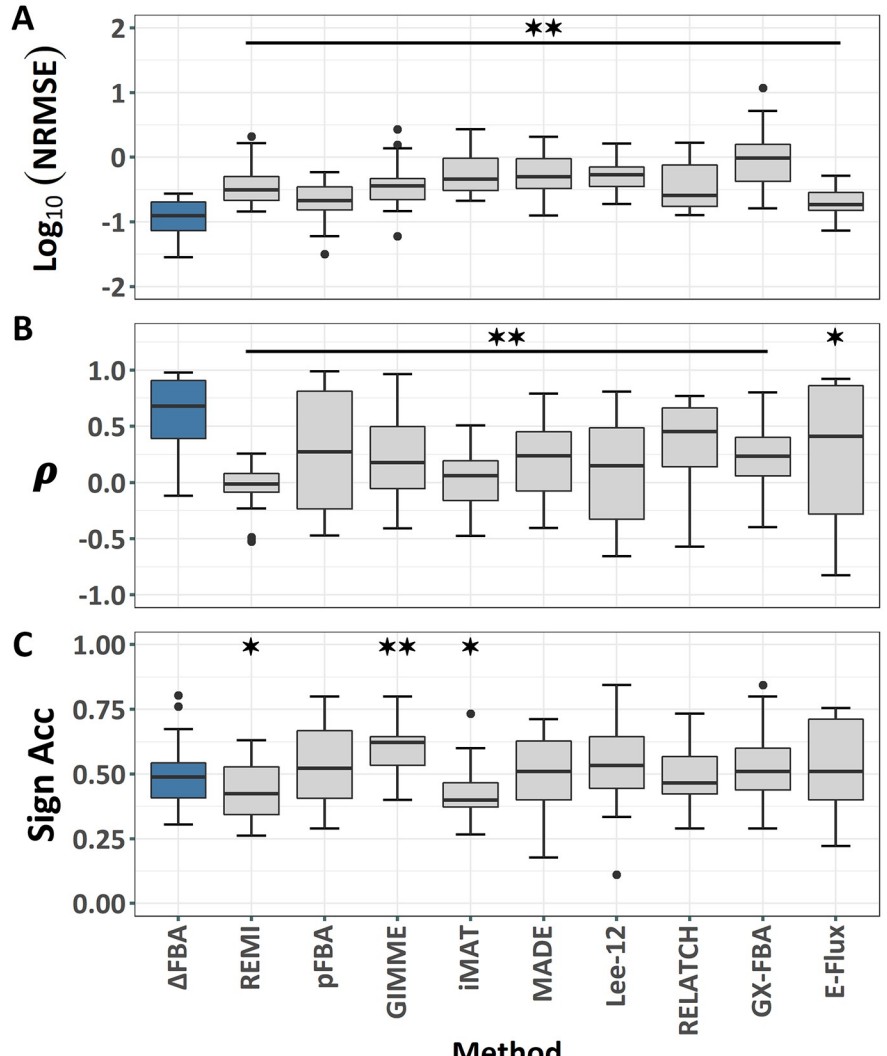

**Fig 1. Comparison of the performance of ΔFBA and 9 comparative FBA methods, including REMI [31], parsimonious FBA (pFBA) [19], GIMME [20], iMAT [21], MADE [22], E-Flux [23], Lee et al. [24], RELATCH [25], and GX-FBA [26], in predicting *E. coli* metabolic response to environmental (dilution rates) and genetic (single gene deletions) perturbations in the Ishii et al. [35] study.** (A) Normalized Root Mean Square Error (NRMSE) of the predicted flux differences; (B) Uncentered Pearson's Correlation Coefficient (ρ); and (C) Sign Accuracy (Sign Acc) between the predicted and measured flux differences. Statistical significance was done using two-sided paired *t*-test. ✶ indicates *p*-value < 0.05 and ✶✶ indicates *p*-value < 0.01.

the other weight coefficients to their default values of 1. However, we noted that using the default $w_j^D = 1$ for the knocked-out reaction produced the same outcome as using $w_j^D = 10$ in this case study. We compared the predicted flux differences using ΔFBA with the measured differences of 46 metabolic fluxes along the central carbon metabolism by incorporating the enzyme expression obtained from RT-PCR. Fig 1 depicts NRMSE, uncentered Pearson correlations, and sign accuracy of the flux differences from ΔFBA, indicating a good agreement between the prediction and the ground truth. The performance of ΔFBA is robust with respect to the thresholds used in Eqs (4)–(9) (see S1 Text and S1 and S2 Figs) and to the cut-off for differential gene expression in specifying the sets $\mathbf{R}^U$ and $\mathbf{R}^D$ (see S3 Fig). The results of ΔFBA

using the whole-genome gene expression profiles for a subset of perturbation experiments are comparable with those using RT-PCR data (see S4 Fig). The prediction accuracy for individual reactions is given in S5 Fig, demonstrating that metabolic reactions that form futile cycles, including reactions along the glycolysis and citric acid cycle, were associated with higher prediction errors. The difficulty in predicting metabolic fluxes in futile cycles is not surprising since such cycles generate degeneracy in FBA [27].

We compared the prediction accuracy of flux differences by ΔFBA with REMI [31] and eight other FBA methods: parsimonious FBA (pFBA) [19], GIMME [20], iMAT [21], MADE [22], E-Flux [23], Lee et al. [24], RELATCH [25], and GX-FBA [26]. Except for pFBA, all of these methods integrated gene expression data for the flux predictions. As illustrated in Fig 1, ΔFBA outperforms the other methods in predicting the flux differences by having statistically significantly lower NRMSE and higher Pearson correlations. Meanwhile, the sign accuracies for all methods are comparable with each other. We noted that roughly 18% of the measured flux differences are exactly 0, while methods generally do not produce any zero flux differences. Here, GIMME performed better than ΔFBA (and other methods) in sign accuracy while having worse NRMSE and Pearson correlation, since the method is more readily able to produce zero flux differences than ΔFBA (e.g., when a reaction is removed from the metabolic network model [20]).

Another study, carried out by Gerosa *et al.* [36], looked at how *E. coli*'s central carbon metabolism adapts to 8 different carbon sources: acetate, fructose, galactose, glucose, glycerol, gluconate, pyruvate and succinate. The study generated $^{13}$C metabolic flux, metabolite concentration and microarray gene expression data from exponentially growing *E. coli* under each carbon source. The study found that only a small subset of the numerous transcriptome changes translates to notable shifts in the corresponding metabolic fluxes, indicating non-trivial relationships between transcriptional regulations and metabolic fluxes. We applied ΔFBA to predict flux changes between every pair of the carbon sources, treating one as the perturbation and another as the control condition. Fig 2 describes the good agreement between the flux difference predictions by ΔFBA with the measured differences of 34 metabolic fluxes between any two carbon sources, specifically in terms of NRMSE (mean: 0.15), uncentered Pearson correlation (mean: 0.61), and sign accuracy (mean: 0.66). The findings from Ishii *et al.* [35] and Gerosa *et al.* [36] highlight the ability of ΔFBA in accurately predicting metabolic flux alterations using transcriptomic data for both environmental (e.g., dilution rates, carbon sources) and genetic perturbations.

## Dysregulation of skeletal muscle metabolism in type-2 diabetes

In this case study, we looked at metabolic alterations of human muscle using the myocyte GEM *iMyocyte2419* [37] and gene expression datasets from two type-2 diabetes (T2D) studies, one by van Tienen *et al.* [43] and another by Jin *et al.* [44]. The study by van Tienen *et al.* [43] compared long term T2D patients with age-matched cohort, and reported the downregulation of gene expression related to substrate transport into mitochondria, conversion of pyruvate into acetyl-CoA, aspartate-malate shuttle in mitochondria, glycolysis, TCA cycle, and electron transport chain. Similarly, Jin *et al.* [44] reported a significant enrichment of pathways involved the oxidative phosphorylation among the downregulated genes in their T2D cohort when compared to control. Jin *et al.* [44] further identified the transcription factor SRF and its cofactor MKL1 among the top-ranking enriched gene sets with increased expression. But, the correlation between the differential gene expressions in the two studies is only modest. [37]

We applied ΔFBA to predict the flux changes based on the differential gene expressions in each of the two studies above (see Materials and methods). We grouped the reactions based on

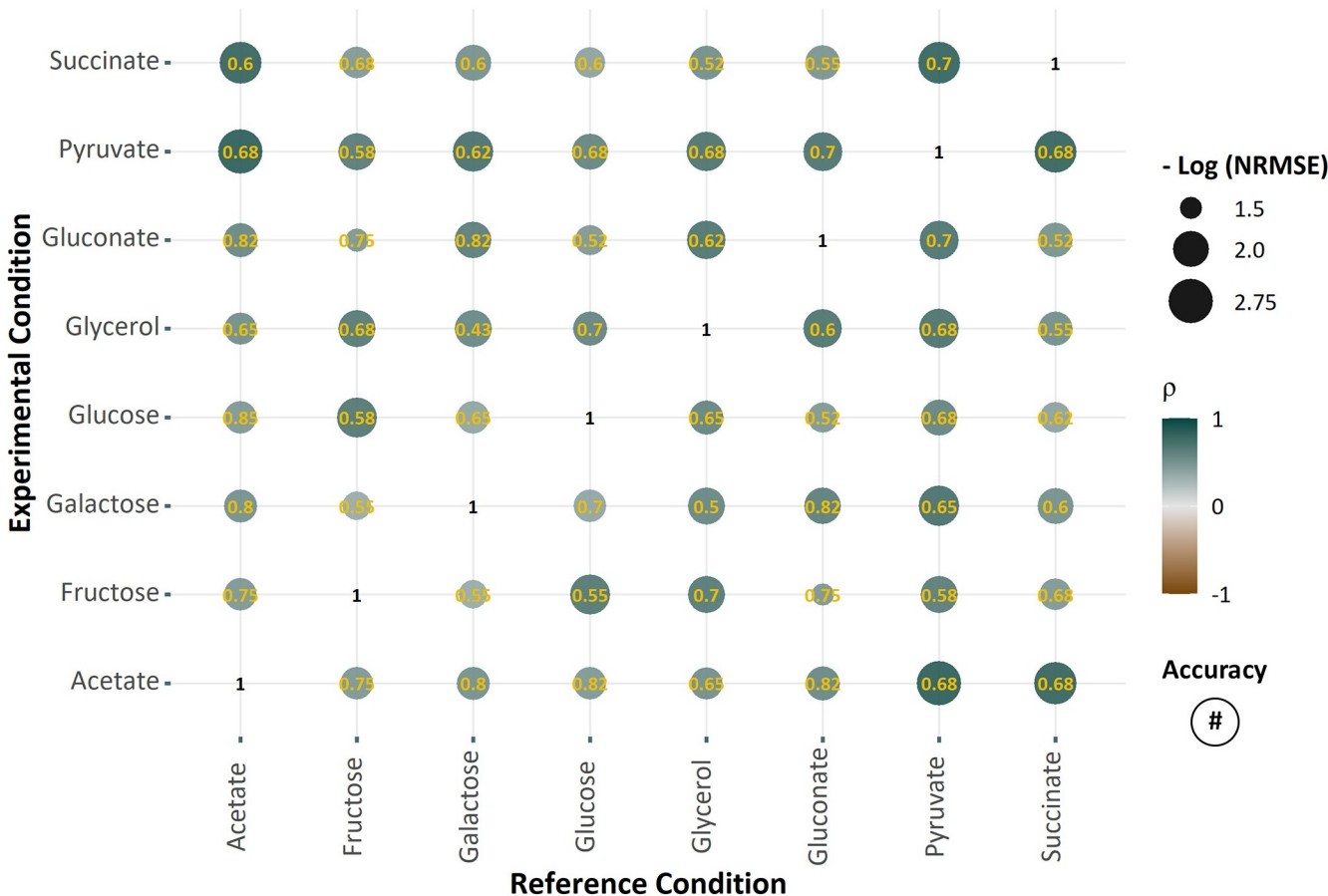

**Fig 2. Prediction of metabolic flux changes in *E. coli* caused by changes in the carbon source using ΔFBA.** The horizontal axis reports the reference carbon source (control) and the vertical axis shows the altered (perturbed) carbon source. Uncentered Pearson's Correlation Coefficient (ρ) is shown by the color of the markers. NRMSE is represented by the size of the markers—the larger the markers, the smaller is the NRMSE. Finally, the directional (sign) accuracy of the flux perturbation predictions is shown by the numbers inside the markers.

whether the predicted flux differences are positive or negative, denoted by up- and down-reactions, respectively. We performed metabolic subsystem enrichment analysis using the subsystems defined in myocyte specific GEM *iMyocyte2419* [37] to identify over-represented metabolic subsystems among the up- and down-reactions (see Materials and methods). As summarized in Fig 3, the enrichment analysis of metabolic changes in the van Tienen *et al.* study shows a significant over-representation of ß-oxidation and BCAA (branched-chain amino acids) metabolism among the down-reactions, and of extracellular transport and lipid metabolism among the up-reactions. The enrichment analysis of flux differences in the Jin *et al.* study also indicates an over-representation of lipid metabolism among the up-reactions in T2D patients, as well as an over-representation of ß-oxidation pathway among the down-reactions (see Fig 3).

Furthermore, we evaluated the difference in the flux throughput for every metabolite irrespective of its compartmental location by computing the difference in the total production flux of each metabolite. Metabolites with a large difference in the flux throughput are of particular interest for disease biomarkers. In the following, we focused on metabolites that have a flux throughput change above a threshold ($|\Delta v_i| > 1\%$ of the largest flux bounds) and excluded

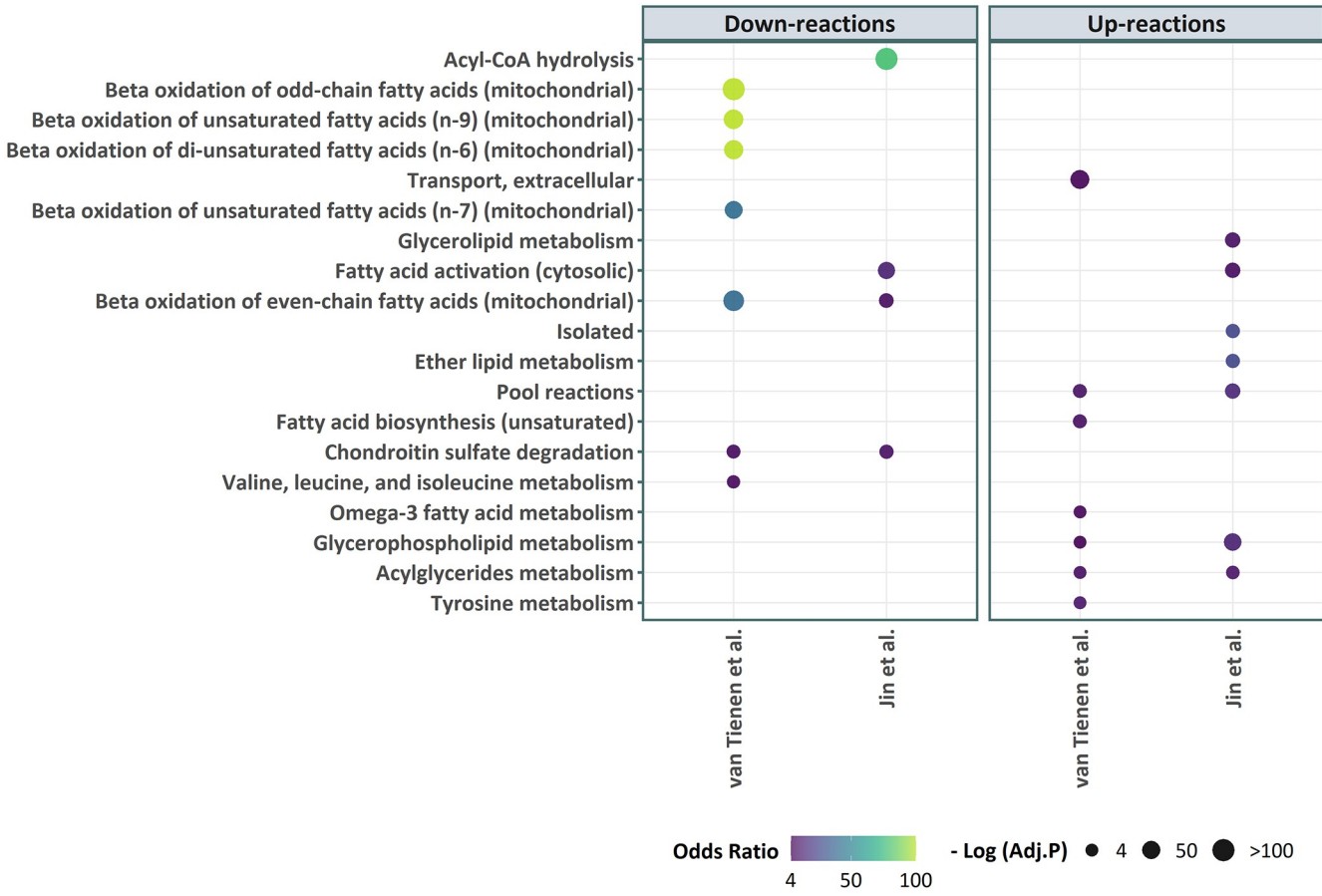

**Fig 3. Enriched metabolic subsystems (FDR<0.05) among the in T2D patients based on flux changes predicted using ΔFBA.** The flux changes were computed based on the transcriptome datasets from two T2D studies: van Tienen *et al.* [43] (GSE19420) and Jin *et al.* [44] (GSE25462). The statistical significance of the over-representation is shown by the size of the markers—larger markers have smaller adjusted *p*-values—while the odds ratio is shown by the color of the markers.

intermediary metabolites that participate in linear reaction sequences. Fig 4 shows the flux throughput differences predicted by ΔFBA for various metabolites. Among the metabolites with a large drop in the flux throughput in both studies are Coenzyme A (CoA), Acetyl-CoA and AMP (Adenosine monophosphate), all of which have been previously identified as metabolite reporters of diabetes [37,46]. Other metabolic biomarkers that have been previously proposed for T2D, such as repression of FAD (Flavin adenine dinucleotide), $FADH_2$ and NADH by van Tienen *et al.* study [43] and increased glycerol by Jin *et al.* study [44], are confirmed by ΔFBA (see Fig 4). Väremo *et al.* [37] had identified these markers of T2D using gene- reaction associations and consensus gene-set analysis in the GEM, *iMyocyte2419*. Besides the above confirmatory observations, ΔFBA results of the two studies further suggest that arachidonate and palmitate are candidate metabolic biomarkers for T2D, both of which have a large positive flux throughput change in the two T2D studies. These metabolites are undetected by simple gene-set analysis using GPR associations in the GEM, but have important roles in the progression and cause of T2D [47–49]. The results above showcase the ability of ΔFBA in elucidating metabolic flux alterations in a complex human GEM and identifying key metabolites of interest in human diseases.

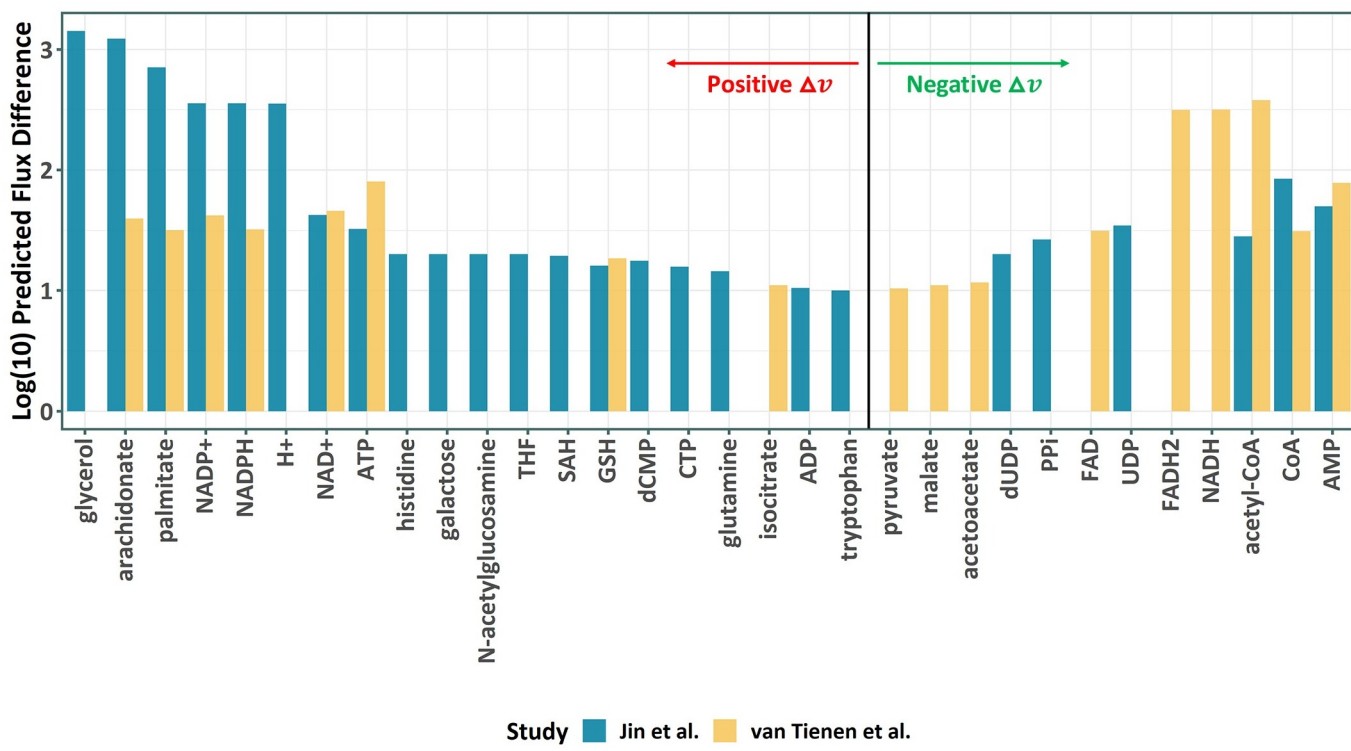

**Fig 4. Alterations in metabolite flux throughput in T2D patients as predicted by ΔFBA.**

## Discussions

GEMs and constraint-based modeling using FBA and the myriad FBA variants have proven to be important enabling tools for establishing genotype-phenotype relationship [10,50,51]. The increasing availability of omics data have driving the development of FBA-based strategies that are able to use such data to improve the accuracy of predictions of intracellular metabolic fluxes. In this work, we present a new FBA-based method, called ΔFBA, built for the purpose of analyzing the metabolic alterations between two conditions given data on differential gene expression. ΔFBA does not require the specification of the metabolic objective, and thus, eliminates any potential pitfalls that are associated with an incorrect selection of this objective. Note that ΔFBA does not generate the flux prediction for a given condition; rather, the method produces differences of metabolic fluxes between two conditions. Differential flux predictions are indispensable in formulating hypothesis and in understanding the physiological response of cells to changes in the environment. ΔFBA can be easily integrated and have been tested to work with the widely popular COBRA toolbox [34].

We showed the applicability and performance of ΔFBA for predicting metabolic flux changes in an array of experimental perturbations and in both simple prokaryotic *E. coli* and complex multicellular human muscle cells. In comparison to other relevant FBA methods, ΔFBA show a markedly better accuracy in prediction metabolic flux changes in *E. coli*. Further, the application of ΔFBA to two T2D studies shed light on the rewiring of muscle metabolism associated with type-2 diabetes that leads to the repression of ß-oxidation and activation of gly-cerolphospholipids, pointing to increased lipid metabolism in the T2D patients. Interestingly, serum metabolic profiling of T2D patients showed increased glycerophospholipids when compared to healthy controls [52]. Besides, clinical and experimental studies have demonstrated

the association between phospholipids and insulin resistance [53]. Furthermore, by looking at the changes in the flux throughput of metabolites, the results of ΔFBA suggest two fatty acids, arachidonate and palmitate, for candidate biomarkers of T2D.

There are several limitations of ΔFBA, the most obvious of which is that the method does not produce flux predictions for individual conditions under comparison. If separate flux predictions for control and perturbed conditions are desired, ΔFBA can be applied synergistically with another FBA method that is capable of predicting single-condition metabolic fluxes. Many of such methods, such as GIMME [20] and iMAT [21], transform gene expression data to a binary state (active/inactive, high/low) and produce metabolic flux prediction for a single condition. But, as shown in Fig 1, pFBA often works as well, if not better, without using gene expression data. When deciding the reference (control) condition, the more well-characterized metabolic state (e.g., more experimental data, more obvious metabolic objective) should be used to generate the reference flux distribution ΔFBA flux differences can be combined with the reference flux values by simple algebra to evaluate metabolic fluxes of the other (perturbed) condition. Such a strategy may be advantageous since once the metabolic flux distribution for the baseline condition is accurately determined (and ideally experimentally validated), one can use ΔFBA and differential gene expression datasets for various perturbation experiments to generate accurate prediction for metabolic fluxes of the perturbed conditions. Note that for many gene expression profiling technologies the relative (differential) expressions are often more reliable and informative of the underlying cellular alterations than the absolute expression because of technical and biological considerations.

Finally, while in the formulation and the application of ΔFBA we considered only differential gene expression data, the method can also accommodate other omics dataset, such as proteomics, by appropriate mapping of the data to changes in reaction expressions. Metabolomics data can also be accommodated in ΔFBA via thermodynamics constraints, as done in REMI [31], in which certain reactions can only proceed in one direction.

## Supporting information

**S1 Text. Threshold criteria for minimum flux change magnitudes.**
(PDF)

**S1 Fig. Comparison of ΔFBA predictions of *E. coli* metabolic response in Ishii *et al*. study [35] using original (relaxed) thresholding in Equation (S1) and stringent thresholding using fold-change reaction expression in Equations (S2)-(S3) (see S1 Text).** (Left) Normalized root mean square error (NRMSE), (Middle) uncentered Pearson's Correlation Coefficient (ρ), (Right) Sign accuracy (Sign Acc) between the predicted flux difference and the measured flux change. The error bars show standard deviation across for 4 dilution rates (0.1, 0.4, 0.5, and 0.7 hours$^{-1}$) and 24 single-gene deletions (*galM*, *glk*, *pgm*, *pgi*, *pfkA*, *pfkB*, *fbp*, *fbaB*, *gapC*, *gpmA*, *gpmB*, *pykA*, *pykF*, *ppsA*, *zwf*, *pgl*, *gnd*, *rpe*, *rpiA*, *rpiB*, *tktA*, *tktB*, *talA*, and *talB*). The difference in performance is not statistically significant.
(TIF)

**S2 Fig. Comparison of FBA performance for different *ε*. Accuracy of ΔFBA predictions of *E. coli* metabolic shifts in response to environmental and genetic perturbations in the Ishii *et al*. study [35].** The default *ε* is 0.1% of the largest flux in the metabolic model under growth maximization and parsimony criteria. The error bars show standard deviation across flux difference predictions for 4 dilution rates and 24 single-gene deletions. The result indicates that the performance of ΔFBA is relatively insensitive to *ε* between 0.01% and 1%.
(TIF)

**S3 Fig. Comparison of ΔFBA performance for different fold-change (FC) expression cut-off for assigning up- and down-regulated reactions (the sets $R^U$ and $R^D$).** The default FC cut-off is 1. The error bars show standard deviation across for 4 dilution rates and 24 single-gene deletions in the Ishii et al. study [35]. The difference in performance is not statistically significant (Mean NRMSE—FC cutoff of 1 = 0.14, FC cutoff of 2 = 0.13; Mean ρ—FC cutoff of 1 = 0.61, FC cutoff of 2 = 0.63; Mean sign accuracy—FC cutoff of 1 = 0.49, FC cutoff of 2 = 0.48).
(TIF)

**S4 Fig. Comparison of ΔFBA performance in predicting *E. coli* metabolic shifts using whole-genome transcriptome data versus using RT-PCR mRNA data.** Directional (Sign Accuracy) agreement and uncentered Pearson's Correlation Coefficient (ρ) between the predicted and measured flux differences have little difference between the incorporation of the two transcriptomic sources using ΔFBA (Mean NRMSE: whole-genome = 0.15, RT-PCR = 0.16; Mean ρ: whole-genome = 0.57; RT-PCR = 0.54; Mean sign accuracy: whole-genome = 0.53, RT-PCR = 0.53).
(TIF)

**S5 Fig. Normalized prediction errors of flux differences by ΔFBA across 46 individual reactions in *E. coli* central carbon metabolism in Ishii *et al*. study [35].** The NRMSE for the full flux differences is shown in blue (leftmost box plot). The remaining box plots in red show the distribution of the normalized error (NE) for each flux *i*: $NE_i = \frac{\sqrt{(\Delta v_i^M - \Delta v_i^*)^2}}{\Delta v_{max}^M - \Delta v_{min}^M}$, across 28 conditions (4 dilution rates and 24 single-gene deletions).
(TIF)

## Acknowledgments

The authors would like to acknowledge the University at Buffalo's Center for Computational Research for computational support.

## Author Contributions

**Conceptualization:** Sudharshan Ravi, Rudiyanto Gunawan.

**Data curation:** Sudharshan Ravi.

**Formal analysis:** Sudharshan Ravi, Rudiyanto Gunawan.

**Funding acquisition:** Rudiyanto Gunawan.

**Investigation:** Sudharshan Ravi, Rudiyanto Gunawan.

**Methodology:** Sudharshan Ravi, Rudiyanto Gunawan.

**Project administration:** Rudiyanto Gunawan.

**Software:** Sudharshan Ravi.

**Supervision:** Rudiyanto Gunawan.

**Validation:** Sudharshan Ravi.

**Visualization:** Sudharshan Ravi, Rudiyanto Gunawan.

**Writing – original draft:** Sudharshan Ravi, Rudiyanto Gunawan.

**Writing – review & editing:** Sudharshan Ravi, Rudiyanto Gunawan.

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
