## [Decision Letter · Decision Letter 0]

6 Apr 2021

Dear Dr. Gunawan,

Thank you very much for submitting your manuscript "ΔFBA - Predicting metabolic flux alterations using genome-scale metabolic models and differential transcriptomic data" for consideration at PLOS Computational Biology.

As with all papers reviewed by the journal, your manuscript was reviewed by members of the editorial board and by several independent reviewers. In light of the reviews (below this email), we would like to invite the resubmission of a significantly-revised version that takes into account the reviewers' comments.

Specifically, while the reviewers appreciate the novelty of the ideas underlying DeltaFBA, they identified several areas in which the study is too preliminary to be considered for publication. This concerns (i) a clear exposition of the technical rationale and details (e.g., regarding mapping from genes to reactions); (ii) demonstration that the method is robust, for, example, with respect to parameter choices; and (iii) demonstration of substantially superior performance compared to a larger selection of state of the art competing methods, based on extended data sets for validation. We appreciate that addressing (ii) and (iii) will likely exceed the normal time frame for re-submission, and we ask you indicate an expected re-submission time.

We cannot make any decision about publication until we have seen the revised manuscript and your response to the reviewers' comments. Your revised manuscript is also likely to be sent to reviewers for further evaluation.

Sincerely,

Joerg Stelling

Associate Editor

PLOS Computational Biology

Jason Papin

Editor-in-Chief

PLOS Computational Biology

Reviewer's Responses to Questions

**Comments to the Authors:**

Reviewer #1: The Authors propose DeltaFBA, a method to identify flux changes using transcriptomic differential data. Although the idea seems novel and timely, there are however some questions that must be addressed:

1) Method Formulation

The section where the MILP problem is presented is not sufficiently clear. Some definitions are lacking, and there are several typos, which hampers the correct interpretation of the proposed method and ultimately may obscure the rationale of DeltaFBA.

The Authors should check that *all* of the variables are formally defined, namely concerning the dimensionality and context. For example:

- L122 - DeltaFBA is not enforcing the equation since S.delta_v=0 derives from considering that each C and P states are stationary.

- L128 – S=0?

- Eq. 1 – max missing? Where is delta_v in the equation?

- The formal definition of z is lacking. Where is z0 in the equation? What is the dimension of z? In Eq.(3), it seems n. That affects the comprehension of the MILP problem.

- The parameters wi are always set to values larger or equal to zero?

2) Choice of the parameters

How can one choose the parameters in an unbiased way? I suppose the values wi, n, vmax, vmin, and epsilon influence the outcome. It would be interesting to more deeply analyze the sensitivity of the results to the choice of these parameters.

3) Correspondence between reactions and enzymes

Does DeltaFBA assume a one-to-one correspondence between reactions and enzymes? It seems that it is the case since the delta_v agrees with the direction of the gene expression changes (L.143), and the sets R^U and R^D are defined a priori. Can the authors comment on what happen when there are many-to-many relationships?

Does the fact that “Only a small fraction of variation in the measured flux ratios can be explained by the fold-change in reaction expressions…” (L.281) affect the overall rationale of the approach? Please comment.

4) Performance evaluation

How are the differences between the measured fluxes and the predicted fluxes distributed? The evaluation measures proposed (L.254) seem not to take individual discrepancies into account; for example, are there any outliers that may be skewing the results? Or, in other words, is it possible to interpret for which fluxes the methods most fail (or are able to better guess the correct value)?

MINOR

Correct “Equation Error! Reference not found” (L.162, L.166).

L.289 – The 4 dilution rates and 24 single-gene deletions are not combinatorial – are the changes combinatorial? Or have they done all-at-once?

Reviewer #2: Ravi and Gunawan present the manuscript title “ΔFBA – Predicting metabolic flux alterations u sing genome-scale metabolic models and differential transcriptomic data” describe an alternative tool to predict metabolic flux alteration by impose additional constraints derived from transcriptome data. The author demonstrated the developed tool and assessed the performance in various experiments of E.coli and human skeleton muscle system.

Comments

- Abstract need to be improve by clearer point out the key contributions of the work.

- There re “Error! Reference source not found” in the manuscript that need to be fixed.

- It is not clear (not be mentioned) how statistical Pvalue which is the important parameter for considering the usage of fold changes. If the fold changes have high pvalues, we cannot use the changes as addition constrains in flux calculation due to uncertainty.

- The gold standard of metabolic flux is C13 labeling experiment which include in some datasets that the author used in case study. I recommend comparing the results from dFBA with C13 flux in details.

- It is not clear for me how performance evaluations were performed. What is the ground truth that author use to estimate performance?

- Based on the report, dFBA have the accuracy <= 0.7, Is it considered good? ROC analysis should be performed and provide in detail for the readers.

- I suggest the authors perform more case studies in yeast system that have a lot of data of transcriptome and C13 flux (some studies provided).

- The manuscript lack of sensitivity analysis that need for new developed tool like dFBA.

- For transparency and reproducibility, the author needs to provide all computational codes used for the case studies presented in the manuscript in the GitHub for the reader.

Reviewer #3: In this work, Ravi and Gunawan present a new method to integrate transcriptome data and genome-scale model to interpret the rewiring in metabolism non-intuitively. While this work is under the scope of PLOS Comp Bio and presents substantially novel and useful algorithm, its superiority and applicability over existing algorithms should be clearly explained than now. Some of my major comments as follows:

1. I think the authors have not thoroughly compared their new method with all the previously existing ones. While they have compared it only with REMI, it is essential to compare it with many other similar algorithms, i.e. integrating differential gene expression data with GEM. While the authors have cited the MOONMIN, Zhu et al as other methods in this category, I’m aware of several others such as tFBA (van Berlo, 2011), MADE (Jensen and Papin 2011), AdaM (Topfer 2012) and GX–FBA (Navid and Elmaas, 2012). Therefore, I would like the authors to comprehensively review the literature in this regard and compare their methods with previous ones, showcasing the strengths of their algorithm over previous ones.

2. In the implementation, it is unclear how the authors deduced up-/down-regulated reactions from up-/down-regulated genes. There exists multiple scenarios which is not straightforward – a reaction with multiple isoenzymes could have different isozymes up-/down-regulated. Similar cases could exist with gene subunits. It is important for authors to provide more details in this regard.

3. Integration of gene expression data with GEM is a one of widely researched topic in this field. So far, two methods have been predominantly proposed – using transcriptome data in the form of differentially expressed genes as used here and the use of a threshold to transform gene expression data onto binary form and then use it. It may be better to authors to comment on this two approaches in their paper and provide how this method complements and/or outperforms the other. I raise this point because, it is well known fact that the transcriptional regulation of metabolic flux is not well correlated and it is good to know which of these methods could capture this well.

4. Several equations in the paper were missing due to formatting error making it further difficult to evaluate the presented algorithm.

5. It is good that the authors have provided the code in github, which helps its reproduction/implementation.

**Have all data underlying the figures and results presented in the manuscript been provided?**

Reviewer #1: Yes

Reviewer #2: Yes

Reviewer #3: Yes

PLOS authors have the option to publish the peer review history of their article (what does this mean?). If published, this will include your full peer review and any attached files.

Reviewer #1: No

Reviewer #2: No

Reviewer #3: **Yes: **Meiyappan Lakshmanan
---

## [Decision Letter · Decision Letter 1]

25 Oct 2021

Dear Dr. Gunawan,

We are pleased to inform you that your manuscript 'ΔFBA - Predicting metabolic flux alterations using genome-scale metabolic models and differential transcriptomic data' has been provisionally accepted for publication in PLOS Computational Biology.

Best regards,

Joerg Stelling

Associate Editor

Jason Papin

Editor-in-Chief

PLOS Computational Biology

Reviewer's Responses to Questions

**Comments to the Authors:**

Reviewer #2: The manuscript was improved.

Reviewer #3: The authors have adequately addressed all my concerns.

**Have the authors made all data and (if applicable) computational code underlying the findings in their manuscript fully available?**

Reviewer #2: None

Reviewer #3: Yes

PLOS authors have the option to publish the peer review history of their article (what does this mean?). If published, this will include your full peer review and any attached files.

Reviewer #2: No

Reviewer #3: **Yes: **Meiyappan Lakshmanan

---

## [Editor Report · Acceptance letter]

5 Nov 2021

PCOMPBIOL-D-21-00222R1 

ΔFBA - Predicting metabolic flux alterations using genome-scale metabolic models and differential transcriptomic data

Dear Dr Gunawan,

I am pleased to inform you that your manuscript has been formally accepted for publication in PLOS Computational Biology. Your manuscript is now with our production department and you will be notified of the publication date in due course.

With kind regards,

Zsofia Freund
